# Bayesian Batch Active Learning as Sparse Subset Approximation

**Robert Pinsler**
Department of Engineering
University of Cambridge
rp586@cam.ac.uk

**Jonathan Gordon**
Department of Engineering
University of Cambridge
jg801@cam.ac.uk

**Eric Nalisnick**
Department of Engineering
University of Cambridge
etn22@cam.ac.uk

**José Miguel Hernández-Lobato**
Department of Engineering
University of Cambridge
jmh233@cam.ac.uk

## Abstract

Leveraging the wealth of unlabeled data produced in recent years provides great potential for improving supervised models. When the cost of acquiring labels is high, probabilistic active learning methods can be used to greedily select the most informative data points to be labeled. However, for many large-scale problems standard greedy procedures become computationally infeasible and suffer from negligible model change. In this paper, we introduce a novel Bayesian batch active learning approach that mitigates these issues. Our approach is motivated by approximating the complete data posterior of the model parameters. While naive batch construction methods result in correlated queries, our algorithm produces diverse batches that enable efficient active learning at scale. We derive interpretable closed-form solutions akin to existing active learning procedures for linear models, and generalize to arbitrary models using random projections. We demonstrate the benefits of our approach on several large-scale regression and classification tasks.

## 1 Introduction

Much of machine learning's success stems from leveraging the wealth of data produced in recent years. However, in many cases expert knowledge is needed to provide labels, and access to these experts is limited by time and cost constraints. For example, cameras could easily provide images of the many fish that inhabit a coral reef, but an ichthyologist would be needed to properly label each fish with the relevant biological information. In such settings, *active learning* (AL) [1] enables data-efficient model training by intelligently selecting points for which labels should be requested.

Taking a Bayesian perspective, a natural approach to AL is to choose the set of points that maximally reduces the uncertainty in the posterior over model parameters [2]. Unfortunately, solving this combinatorial optimization problem is NP-hard. Most AL methods iteratively solve a greedy approximation, e.g. using maximum entropy [3] or maximum information gain [2, 4]. These approaches alternate between querying a single data point and updating the model, until the query budget is exhausted. However, as we discuss below, sequential greedy methods have severe limitations in modern machine learning applications, where datasets are massive and models often have millions of parameters.

A possible remedy is to select an entire *batch* of points at every AL iteration. Batch AL approaches dramatically reduce the computational burden caused by repeated model updates, while resulting in much more significant learning updates. It is also more practical in applications where the cost of

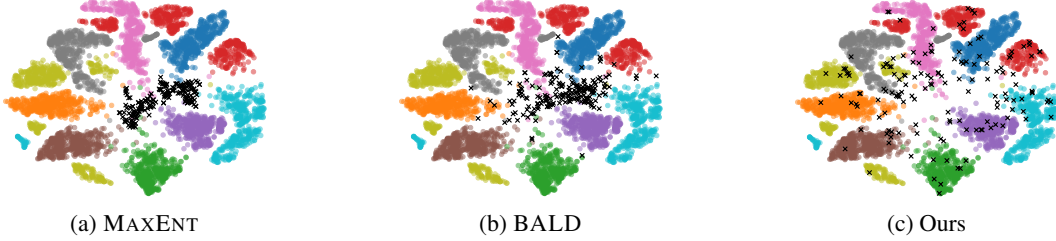

|                    |                    |              |
|:------------------:|:------------------:|:------------:|
| (a) MAXENT         | (b) BALD           | (c) Ours     |

Figure 1: Batch construction of different AL methods on *cifar10*, shown as a t-SNE projection [12]. Given 5000 labeled points (colored by class), a batch of 200 points (black crosses) is queried.

acquiring labels is high but can be parallelized. Examples include crowd-sourcing a complex labeling task, leveraging parallel simulations on a compute cluster, or performing experiments that require resources with time-limited availability (e.g. a wet-lab in natural sciences). Unfortunately, naively constructing a batch using traditional acquisition functions still leads to highly correlated queries [5], i.e. a large part of the budget is spent on repeatedly choosing nearby points. Despite recent interest in batch methods [5–8], there currently exists no principled, scalable Bayesian batch AL algorithm.

In this paper, we propose a novel Bayesian batch AL approach that mitigates these issues. The key idea is to re-cast batch construction as optimizing a sparse subset approximation to the log posterior induced by the full dataset. This formulation of AL is inspired by recent work on *Bayesian coresets* [9, 10]. We leverage these similarities and use the Frank-Wolfe algorithm [11] to enable efficient Bayesian AL at scale. We derive interpretable closed-form solutions for linear and probit regression models, revealing close connections to existing AL methods in these cases. By using random projections, we further generalize our algorithm to work with any model with a tractable likelihood. We demonstrate the benefits of our approach on several large-scale regression and classification tasks.

## 2 Background

We consider discriminative models $p(\boldsymbol{y}|\boldsymbol{x}, \boldsymbol{\theta})$ parameterized by $\boldsymbol{\theta} \in \Theta$, mapping from inputs $\boldsymbol{x} \in \mathcal{X}$ to a distribution over outputs $\boldsymbol{y} \in \mathcal{Y}$. Given a labeled dataset $\mathcal{D}_0 = \{\boldsymbol{x}_n, \boldsymbol{y}_n\}_{n=1}^N$, the learning task consists of performing inference over the parameters $\boldsymbol{\theta}$ to obtain the posterior distribution $p(\boldsymbol{\theta}|\mathcal{D}_0)$. In the AL setting [1], the learner is allowed to choose the data points from which it learns. In addition to the initial dataset $\mathcal{D}_0$, we assume access to (i) an unlabeled pool set $\mathcal{X}_p = \{\boldsymbol{x}_m\}_{m=1}^M$, and (ii) an oracle labeling mechanism which can provide labels $\mathcal{Y}_p = \{\boldsymbol{y}_m\}_{m=1}^M$ for the corresponding inputs.

Probabilistic AL approaches choose points by considering the posterior distribution of the model parameters. Without any budget constraints, we could query the oracle $M$ times, yielding the complete data posterior through Bayes' rule,

$$p(\boldsymbol{\theta}|\mathcal{D}_0 \cup (\mathcal{X}_p, \mathcal{Y}_p)) = \frac{p(\boldsymbol{\theta}|\mathcal{D}_0)\, p(\mathcal{Y}_p|\mathcal{X}_p, \boldsymbol{\theta})}{p(\mathcal{Y}_p|\mathcal{X}_p, \mathcal{D}_0)}, \tag{1}$$

where here $p(\boldsymbol{\theta}|\mathcal{D}_0)$ plays the role of the prior. While the complete data posterior is optimal from a Bayesian perspective, in practice we can only select a subset, or batch, of points $\mathcal{D}' = (\mathcal{X}', \mathcal{Y}') \subseteq \mathcal{D}_p$ due to budget constraints. From an information-theoretic perspective [2], we want to query points $\mathcal{X}' \subseteq \mathcal{X}_p$ that are *maximally informative*, i.e. minimize the *expected* posterior entropy,

$$\mathcal{X}^* = \underset{\mathcal{X}' \subseteq \mathcal{X}_p, |\mathcal{X}'| \leq b}{\arg\min} \mathbb{E}_{\mathcal{Y}' \sim p(\mathcal{Y}'|\mathcal{X}', \mathcal{D}_0)} \left[ \mathbb{H}\left[ \boldsymbol{\theta}|\mathcal{D}_0 \cup (\mathcal{X}', \mathcal{Y}') \right] \right], \tag{2}$$

where $b$ is a query budget. Solving Eq. (2) directly is intractable, as it requires considering all possible subsets of the pool set. As such, most AL strategies follow a myopic approach that iteratively chooses a single point until the budget is exhausted. Simple heuristics, e.g. maximizing the predictive entropy (MAXENT), are often employed [13, 5]. Houlsby et al. [4] propose BALD, a greedy approximation to Eq. (2) which seeks the point $\boldsymbol{x}$ that maximizes the decrease in expected entropy:

$$\boldsymbol{x}^* = \underset{\boldsymbol{x} \in \mathcal{X}_p}{\arg\min} \; \mathbb{H}\left[ \boldsymbol{\theta}|\mathcal{D}_0 \right] - \mathbb{E}_{\boldsymbol{y} \sim p(\boldsymbol{y}|\boldsymbol{x}, \mathcal{D}_0)} \left[ \mathbb{H}\left[ \boldsymbol{\theta}|\boldsymbol{x}, \boldsymbol{y}, \mathcal{D}_0 \right] \right]. \tag{3}$$

While sequential greedy strategies can be near-optimal in certain cases [14, 15], they become severely limited for large-scale settings. In particular, it is computationally infeasible to re-train the model

after every acquired data point, e.g. re-training a ResNet [16] thousands of times is clearly impractical. Even if such an approach were feasible, the addition of a single point to the training set is likely to have a negligible effect on the parameter posterior distribution [5]. Since the model changes only marginally after each update, subsequent queries thus result in acquiring similar points in data space. As a consequence, there has been renewed interest in finding tractable batch AL formulations. Perhaps the simplest approach is to naively select the $b$ highest-scoring points according to a standard acquisition function. However, such naive batch construction methods still result in highly correlated queries [5]. This issue is highlighted in Fig. 1, where both MAXENT (Fig. 1a) and BALD (Fig. 1b) expend a large part of the budget on repeatedly choosing nearby points.

## 3  Bayesian batch active learning as sparse subset approximation

We propose a novel probabilistic batch AL algorithm that mitigates the issues mentioned above. Our method generates batches that cover the entire data manifold (Fig. 1c), and, as we will show later, are highly effective for performing posterior inference over the model parameters. Note that while our approach alternates between acquiring data points and updating the model for several iterations in practice, we restrict the derivations hereafter to a single iteration for simplicity.

The key idea behind our batch AL approach is to choose a batch $\mathcal{D}'$, such that the updated log posterior $\log p(\boldsymbol{\theta}|\mathcal{D}_0 \cup \mathcal{D}')$ best approximates the complete data log posterior $\log p(\boldsymbol{\theta}|\mathcal{D}_0 \cup \mathcal{D}_p)$. In AL, we do not have access to the labels before querying the pool set. We therefore take expectation w.r.t. the current predictive posterior distribution $p(\mathcal{Y}_p|\mathcal{X}_p, \mathcal{D}_0) = \int p(\mathcal{Y}_p|\mathcal{X}_p, \boldsymbol{\theta}) \, p(\boldsymbol{\theta}|\mathcal{D}_0) d\boldsymbol{\theta}$. The *expected* complete data log posterior is thus

$$
\begin{aligned}
\mathop{\mathbb{E}}_{\mathcal{Y}_p} \left[ \log p(\boldsymbol{\theta}|\mathcal{D}_0 \cup (\mathcal{X}_p, \mathcal{Y}_p)) \right] &= \mathop{\mathbb{E}}_{\mathcal{Y}_p} \left[ \log p(\boldsymbol{\theta}|\mathcal{D}_0) + \log p(\mathcal{Y}_p|\mathcal{X}_p, \boldsymbol{\theta}) - \log p(\mathcal{Y}_p|\mathcal{X}_p, \mathcal{D}_0) \right] \\
&= \log p(\boldsymbol{\theta}|\mathcal{D}_0) + \mathop{\mathbb{E}}_{\mathcal{Y}_p} \left[ \log p(\mathcal{Y}_p|\mathcal{X}_p, \boldsymbol{\theta}) \right] + \mathbb{H}[\mathcal{Y}_p|\mathcal{X}_p, \mathcal{D}_0] \\
&= \log p(\boldsymbol{\theta}|\mathcal{D}_0) + \sum_{m=1}^{M} \Bigg( \underbrace{\mathop{\mathbb{E}}_{\boldsymbol{y}_m} \left[ \log p(\boldsymbol{y}_m|\boldsymbol{x}_m, \boldsymbol{\theta}) \right] + \mathbb{H} \left[ \boldsymbol{y}_m|\boldsymbol{x}_m, \mathcal{D}_0 \right]}_{\mathcal{L}_m(\boldsymbol{\theta})} \Bigg),
\end{aligned}
\tag{4}
$$

where the first equality uses Bayes' rule (cf. Eq. (1)), and the third equality assumes conditional independence of the outputs given the inputs. This assumption holds for the type of factorized predictive posteriors we consider, e.g. as induced by Gaussian or Multinomial likelihood models.

**Batch construction as sparse approximation**   Taking inspiration from Bayesian coresets [9, 10], we re-cast Bayesian batch construction as a sparse approximation to the expected complete data log posterior. Since the first term in Eq. (4) only depends on $\mathcal{D}_0$, it suffices to choose the batch that best approximates $\sum_m \mathcal{L}_m(\boldsymbol{\theta})$. Similar to Campbell and Broderick [10], we view $\mathcal{L}_m : \Theta \mapsto \mathbb{R}$ and $\mathcal{L} = \sum_m \mathcal{L}_m$ as vectors in function space. Letting $\boldsymbol{w} \in \{0, 1\}^M$ be a weight vector indicating which points to include in the AL batch, and denoting $\mathcal{L}(\boldsymbol{w}) = \sum_m w_m \mathcal{L}_m$ (with slight abuse of notation), we convert the problem of constructing a batch to a sparse subset approximation problem, i.e.

$$
\boldsymbol{w}^* = \operatorname*{minimize}_{\boldsymbol{w}} \|\mathcal{L} - \mathcal{L}(\boldsymbol{w})\|^2 \quad \text{subject to} \quad w_m \in \{0, 1\} \quad \forall m, \ \sum_m \mathbb{1}_m \leq b.
\tag{5}
$$

Intuitively, Eq. (5) captures the key objective of our framework: a "good" approximation to $\mathcal{L}$ implies that the resulting posterior will be close to the (expected) posterior had we observed the complete pool set. Since solving Eq. (5) is generally intractable, in what follows we propose a generic algorithm to efficiently find an approximate solution.

**Inner products and Hilbert spaces**   We propose to construct our batches by solving Eq. (5) in a Hilbert space induced by an inner product $\langle \mathcal{L}_n, \mathcal{L}_m \rangle$ between function vectors, with associated norm $\| \cdot \|$. Below, we discuss the choice of specific inner products. Importantly, this choice introduces a notion of *directionality* into the optimization procedure, enabling our approach to adaptively construct query batches while implicitly accounting for similarity between selected points.

**Frank-Wolfe optimization** To approximately solve the optimization problem in Eq. (5) we follow the work of Campbell and Broderick [10], i.e. we relax the binary weight constraint to be non-negative and replace the cardinality constraint with a polytope constraint. Let $\sigma_m = \|\mathcal{L}_m\|$, $\sigma = \sum_m \sigma_m$, and $\boldsymbol{K} \in \mathbb{R}^{M \times M}$ be a kernel matrix with $K_{mn} = \langle \mathcal{L}_m, \mathcal{L}_n \rangle$. The relaxed optimization problem is

$$\underset{\boldsymbol{w}}{\text{minimize}} \ (\mathbf{1} - \boldsymbol{w})^T \boldsymbol{K} (\mathbf{1} - \boldsymbol{w}) \quad \text{subject to} \quad w_m \geq 0 \quad \forall m, \ \sum_m w_m \sigma_m = \sigma, \tag{6}$$

where we used $\|\mathcal{L} - \mathcal{L}(\boldsymbol{w})\|^2 = (\mathbf{1} - \boldsymbol{w})^T \boldsymbol{K} (\mathbf{1} - \boldsymbol{w})$. The polytope has vertices $\{\sigma/\sigma_m \, \mathbf{1}_m\}_{m=1}^M$ and contains the point $\boldsymbol{w} = [1, 1, \ldots, 1]^T$. Eq. (6) can be solved efficiently using the Frank-Wolfe algorithm [11], yielding the optimal weights $\boldsymbol{w}^*$ after $b$ iterations. The complete AL procedure, *Active Bayesian CoreSets with Frank-Wolfe optimization* (ACS-FW), is outlined in Appendix A (see Algorithm A.1). The key computation in Algorithm A.1 (Line 6) is

$$\left\langle \mathcal{L} - \mathcal{L}(\boldsymbol{w}), \frac{1}{\sigma_n} \mathcal{L}_n \right\rangle = \frac{1}{\sigma_n} \sum_{m=1}^N (1 - w_m) \langle \mathcal{L}_m, \mathcal{L}_n \rangle, \tag{7}$$

which only depends on the inner products $\langle \mathcal{L}_m, \mathcal{L}_n \rangle$ and norms $\sigma_n = \|\mathcal{L}_n\|$. At each iteration, the algorithm greedily selects the vector $\mathcal{L}_f$ most aligned with the residual error $\mathcal{L} - \mathcal{L}(\boldsymbol{w})$. The weights $\boldsymbol{w}$ are then updated according to a line search along the $f^{\text{th}}$ vertex of the polytope (recall that the optimum of a convex objective over a polytope—as in Eq. (6)—is attained at the vertices), which by construction is the $f^{\text{th}}$-coordinate unit vector. This corresponds to adding at most one data point to the batch in every iteration. Since the algorithm allows to re-select indices from previous iterations, the resulting weight vector has $\leq b$ non-zero entries. Empirically, we find that this property leads to smaller batches as more data points are acquired.

Since it is non-trivial to leverage the continuous weights returned by the Frank-Wolfe algorithm in a principled way, the final step of our algorithm is to project the weights back to the feasible space, i.e. set $\tilde{w}_m^* = 1$ if $w_m^* > 0$, and 0 otherwise. While this projection step increases the approximation error, we show in Section 7 that our method is still effective in practice. We leave the exploration of alternative optimization procedures that do not require this projection step to future work.

**Choice of inner products** We employ weighted inner products of the form $\langle \mathcal{L}_n, \mathcal{L}_m \rangle_{\hat{\pi}} = \mathbb{E}_{\hat{\pi}} [\langle \mathcal{L}_n, \mathcal{L}_m \rangle]$, where we choose $\hat{\pi}$ to be the current posterior $p(\boldsymbol{\theta}|\mathcal{D}_0)$. We consider two specific inner products with desirable analytical and computational properties; however, other choices are possible. First, we define the weighted Fisher inner product [17, 10]

$$\langle \mathcal{L}_n, \mathcal{L}_m \rangle_{\hat{\pi}, \mathcal{F}} = \underset{\hat{\pi}}{\mathbb{E}} \left[ \nabla_{\boldsymbol{\theta}} \mathcal{L}_n(\boldsymbol{\theta})^T \nabla_{\boldsymbol{\theta}} \mathcal{L}_m(\boldsymbol{\theta}) \right], \tag{8}$$

which is reminiscent of information-theoretic quantities but requires taking gradients of the expected log-likelihood terms[1] w.r.t. the parameters. In Section 4, we show that for specific models this choice leads to simple, interpretable expressions that are closely related to existing AL procedures.

An alternative choice that lifts the restriction of having to compute gradients is the weighted Euclidean inner product, which considers the marginal likelihood of data points [10],

$$\langle \mathcal{L}_n, \mathcal{L}_m \rangle_{\hat{\pi}, 2} = \underset{\hat{\pi}}{\mathbb{E}} \left[ \mathcal{L}_n(\boldsymbol{\theta}) \mathcal{L}_m(\boldsymbol{\theta}) \right]. \tag{9}$$

The key advantage of this inner product is that it only requires tractable likelihood computations. In Section 5 this will prove highly useful in providing a black-box method for these computations in any model (that has a tractable likelihood) using random feature projections.

**Method overview** In summary, we (i) consider the $\mathcal{L}_m$ in Eq. (4) as vectors in function space and re-cast batch construction as a sparse approximation to the full data log posterior from Eq. (5); (ii) replace the cardinality constraint with a polytope constraint in a Hilbert space, and relax the binary weight constraint to non-negativity; (iii) solve the resulting optimization problem in Eq. (6) using Algorithm A.1; (iv) construct the AL batch by including all points $\boldsymbol{x}_m \in \mathcal{X}_p$ with $w_m^* > 0$.

# 4   Analytic expressions for linear models

In this section, we use the weighted Fisher inner product from Eq. (8) to derive closed-form expressions of the key quantities of our algorithm for two types of models: Bayesian linear regression and probit regression. Although the considered models are relatively simple, they can be used flexibly to construct more powerful models that still admit closed-form solutions. For example, in Section 7 we demonstrate how using neural linear models [18, 19] allows to perform efficient AL on several regression tasks. We consider arbitrary models and inference procedures in Section 5.

**Linear regression**   Consider the following model for scalar Bayesian linear regression,

$$y_n = \boldsymbol{\theta}^T \boldsymbol{x}_n + \epsilon_n, \quad \epsilon_n \sim \mathcal{N}(0, \sigma_0^2), \quad \boldsymbol{\theta} \sim p(\boldsymbol{\theta}), \tag{10}$$

where $p(\boldsymbol{\theta})$ is a factorized Gaussian prior with unit variance; extensions to richer Gaussian priors are straightforward. Given a labeled dataset $\mathcal{D}_0$, the posterior is given in closed form as $p(\boldsymbol{\theta}|D_0, \sigma_0^2) = \mathcal{N}\left(\boldsymbol{\theta}; (\boldsymbol{X}^T\boldsymbol{X} + \sigma_0^2\boldsymbol{I})^{-1}\boldsymbol{X}^T\boldsymbol{y}, \boldsymbol{\Sigma_\theta}\right)$ with $\boldsymbol{\Sigma_\theta} = \sigma_0^2(\boldsymbol{X}^T\boldsymbol{X} + \sigma_0^2\boldsymbol{I})^{-1}$. For this model, a closed-form expression for the inner product in Eq. (8) is

$$\langle \mathcal{L}_n, \mathcal{L}_m \rangle_{\hat{\pi}, \mathcal{F}} = \frac{\boldsymbol{x}_n^T \boldsymbol{x}_m}{\sigma_0^4} \boldsymbol{x}_n^T \boldsymbol{\Sigma_\theta} \boldsymbol{x}_m, \tag{11}$$

where $\hat{\pi}$ is chosen to be the posterior $p(\boldsymbol{\theta}|\mathcal{D}_0, \sigma_0^2)$. See Appendix B.1 for details on this derivation. We can make a direct comparison with BALD [2, 4] by treating the squared norm of a data point with itself as a greedy acquisition function,[2] $\alpha_{\text{ACS}}(\boldsymbol{x}_n; \mathcal{D}_0) = \langle \mathcal{L}_n, \mathcal{L}_n \rangle_{\hat{\pi}, \mathcal{F}}$, yielding,

$$\alpha_{\text{ACS}}(\boldsymbol{x}_n; \mathcal{D}_0) = \frac{\boldsymbol{x}_n^T \boldsymbol{x}_n}{\sigma_0^4} \boldsymbol{x}_n^T \boldsymbol{\Sigma_\theta} \boldsymbol{x}_n, \qquad \alpha_{\text{BALD}}(\boldsymbol{x}_n; \mathcal{D}_0) = \frac{1}{2} \log\left(1 + \frac{\boldsymbol{x}_n^T \boldsymbol{\Sigma_\theta} \boldsymbol{x}_n}{\sigma_0^2}\right). \tag{12}$$

The two functions share the term $\boldsymbol{x}_n^T \boldsymbol{\Sigma_\theta} \boldsymbol{x}_n$, but BALD wraps the term in a logarithm whereas $\alpha_{\text{ACS}}$ scales it by $\boldsymbol{x}_n^T \boldsymbol{x}_n$. Ignoring the $\boldsymbol{x}_n^T \boldsymbol{x}_n$ term in $\alpha_{\text{ACS}}$ makes the two quantities proportional—$\exp(2\alpha_{\text{BALD}}(\boldsymbol{x}_n; \mathcal{D}_0)) \propto \alpha_{\text{ACS}}(\boldsymbol{x}_n; \mathcal{D}_0)$—and thus equivalent under a greedy maximizer. Another observation is that $\boldsymbol{x}_n^T \boldsymbol{\Sigma_\theta} \boldsymbol{x}_n$ is very similar to a *leverage score* [20–22], which is computed as $\boldsymbol{x}_n^T (\boldsymbol{X}^T\boldsymbol{X})^{-1} \boldsymbol{x}_n$ and quantifies the degree to which $\boldsymbol{x}_n$ influences the least-squares solution. We can then interpret the $\boldsymbol{x}_n^T \boldsymbol{x}_n$ term in $\alpha_{\text{ACS}}$ as allowing for more contribution from the current instance $\boldsymbol{x}_n$ than BALD or leverage scores would.

**Probit regression**   Consider the following model for Bayesian probit regression,

$$p(y_n|\boldsymbol{x}_n, \boldsymbol{\theta}) = \text{Ber}\left(\Phi(\boldsymbol{\theta}^T \boldsymbol{x}_n)\right), \quad \boldsymbol{\theta} \sim p(\boldsymbol{\theta}), \tag{13}$$

where $\Phi(\cdot)$ represents the standard Normal cumulative density function (cdf), and $p(\boldsymbol{\theta})$ is assumed to be a factorized Gaussian with unit variance. We obtain a closed-form solution for Eq. (8), i.e.

$$\langle \mathcal{L}_n, \mathcal{L}_m \rangle_{\hat{\pi}, \mathcal{F}} = \boldsymbol{x}_n^T \boldsymbol{x}_m \Big( \text{BvN}\left(\zeta_n, \zeta_m, \rho_{n,m}\right) - \Phi(\zeta_n)\Phi(\zeta_m) \Big) \tag{14}$$

$$\zeta_i = \frac{\boldsymbol{\mu_\theta}^T \boldsymbol{x}_i}{\sqrt{1 + \boldsymbol{x}_i^T \boldsymbol{\Sigma_\theta} \boldsymbol{x}_i}} \qquad \rho_{n,m} = \frac{\boldsymbol{x}_n^T \boldsymbol{\Sigma_\theta} \boldsymbol{x}_m}{\sqrt{1 + \boldsymbol{x}_n^T \boldsymbol{\Sigma_\theta} \boldsymbol{x}_n}\sqrt{1 + \boldsymbol{x}_m^T \boldsymbol{\Sigma_\theta} \boldsymbol{x}_m}},$$

where $\text{BvN}(\cdot)$ is the bi-variate Normal cdf. We again view $\alpha_{\text{ACS}}(\boldsymbol{x}_n; \mathcal{D}_0) = \langle \mathcal{L}_n, \mathcal{L}_n \rangle_{\hat{\pi}, \mathcal{F}}$ as an acquisition function and re-write Eq. (14) as

$$\alpha_{\text{ACS}}(\boldsymbol{x}_n; \mathcal{D}_0) = \boldsymbol{x}_n^T \boldsymbol{x}_n \left( \Phi\left(\zeta_n\right)\left(1 - \Phi\left(\zeta_n\right)\right) - 2\text{T}\left(\zeta_n, \frac{1}{\sqrt{1 + 2\boldsymbol{x}_n^T \boldsymbol{\Sigma_\theta} \boldsymbol{x}_n}}\right)\right), \tag{15}$$

where $\text{T}(\cdot, \cdot)$ is Owen's T function [23]. See Appendix B.2 for the full derivation of Eqs. (14) and (15). Eq. (15) has a simple and intuitive form that accounts for the magnitude of the input vector and a regularized term for the predictive variance.

# 5 Random projections for non-linear models

In Section 4, we have derived closed-form expressions of the weighted Fisher inner product for two specific types of models. However, this approach suffers from two shortcomings. First, it is limited to models for which the inner product can be evaluated in closed form, e.g. linear regression or probit regression. Second, the resulting algorithm requires $\mathcal{O}\left(|\mathcal{P}|^2\right)$ computations to construct a batch, restricting our approach to moderately-sized pool sets.

We address both of these issues using random feature projections, allowing us to approximate the key quantities required for the batch construction. In Algorithm A.2, we introduce a procedure that works for *any* model with a tractable likelihood, scaling only linearly in the pool set size $|\mathcal{P}|$. To keep the exposition simple, we consider models in which the expectation of $\mathcal{L}_n(\boldsymbol{\theta})$ w.r.t. $p(\boldsymbol{y}_n|\boldsymbol{x}_n, \mathcal{D}_0)$ is tractable, but we stress that our algorithm could work with sampling for that expectation as well.

While it is easy to construct a projection for the weighted Fisher inner product [10], its dependence on the number of model parameters through the gradient makes it difficult to scale it to more complex models. We therefore only consider projections for the weighted Euclidean inner product from Eq. (9), which we found to perform comparably in practice. The appropriate projection is [10]

$$\hat{\mathcal{L}}_n = \frac{1}{\sqrt{J}} \left[\mathcal{L}_n(\boldsymbol{\theta}_1), \cdots, \mathcal{L}_n(\boldsymbol{\theta}_J)\right]^T, \qquad \boldsymbol{\theta}_j \sim \hat{\pi}, \tag{16}$$

i.e. $\hat{\mathcal{L}}_n$ represents the $J$-dimensional projection of $\mathcal{L}_n$ in Euclidean space. Given this projection, we are able to approximate inner products as dot products between vectors,

$$\langle \mathcal{L}_n, \mathcal{L}_m \rangle_{\hat{\pi},2} \approx \hat{\mathcal{L}}_n^T \hat{\mathcal{L}}_m, \tag{17}$$

where $\hat{\mathcal{L}}_n^T \hat{\mathcal{L}}_m$ can be viewed as an unbiased sample estimator of $\langle \mathcal{L}_n, \mathcal{L}_m \rangle_{\hat{\pi},2}$ using $J$ Monte Carlo samples from the posterior $\hat{\pi}$. Importantly, Eq. (16) can be calculated for any model with a tractable likelihood. Since in practice we only require inner products of the form $\langle \mathcal{L} - \mathcal{L}(\boldsymbol{w}), \mathcal{L}_n/\sigma_n \rangle_{\hat{\pi},2}$, batches can be efficiently constructed in $\mathcal{O}(|\mathcal{P}|J)$ time. As we show in Section 7, this enables us to scale our algorithm up to pool sets comprising hundreds of thousands of examples.

# 6 Related work

Bayesian AL approaches attempt to query points that maximally reduce model uncertainty. Common heuristics to this intractable problem greedily choose points where the predictive posterior is most uncertain, e.g. maximum variance and maximum entropy [3], or that maximally improve the expected information gain [2, 4]. Scaling these methods to the batch setting in a principled way is difficult for complex, non-linear models. Recent work on improving inference for AL with deep probabilistic models [24, 13] used datasets with at most 10 000 data points and few model updates.

Consequently, there has been great interest in batch AL recently. The literature is dominated by non-probabilistic methods, which commonly trade off diversity and uncertainty. Many approaches are model-specific, e.g. for linear regression [25], logistic regression [26, 27], and k-nearest neighbors [28]; our method works for any model with a tractable likelihood. Others [6–8] follow optimization-based approaches that require optimization over a large number of variables. As these methods scale quadratically with the number of data points, they are limited to smaller pool sets.

Probabilistic batch methods mostly focus on Bayesian optimization problems. Several approaches select the batch that jointly optimizes the acquisition function [29, 30]. As they scale poorly with the batch size, greedy batch construction algorithms are often used instead [31–34]. A common strategy is to impute the labels of the selected data points and update the model accordingly [33]. Our approach also uses the model to predict the labels, but importantly it does not require to update the model after every data point. Moreover, most of the methods in Bayesian optimization employ Gaussian process models. While AL with non-parametric models [35] could benefit from that work, scaling such models to large datasets remains challenging. Our work therefore provides the first principled, scalable and model-agnostic Bayesian batch AL approach.

Similar to us, Sener and Savarese [5] formulate AL as a core-set selection problem. They construct batches by solving a $k$-center problem, attempting to minimize the maximum distance to one of the $k$ queried data points. Since this approach heavily relies on the geometry in data space, it requires

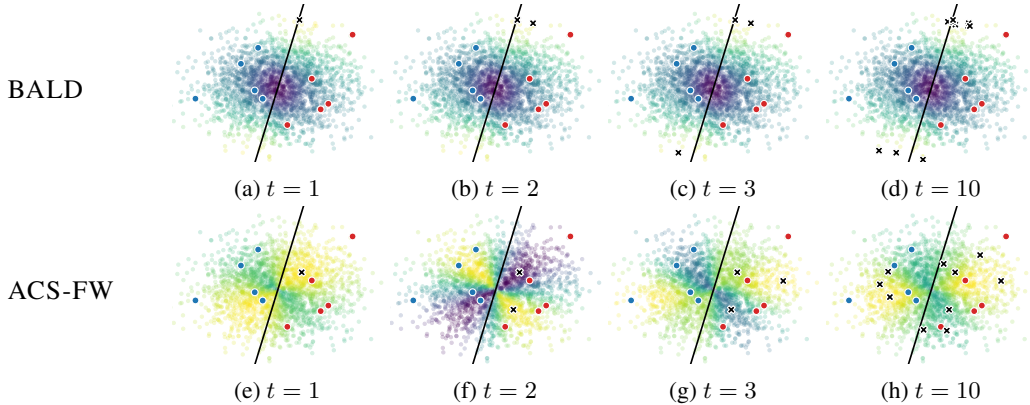

BALD

(a) $t = 1$      (b) $t = 2$      (c) $t = 3$      (d) $t = 10$

ACS-FW

(e) $t = 1$      (f) $t = 2$      (g) $t = 3$      (h) $t = 10$

Figure 2: Batches constructed by BALD (top) and ACS-FW (bottom) on a probit regression task. 10 training data points (red, blue) were sampled from a standard bi-variate Normal, and labeled according to $p(y|\boldsymbol{x}) = \mathrm{Ber}(5x_1 + 0x_2)$. At each step $t$, one unlabeled point (black cross) is queried from the pool set (colored according to acquisition function[4]; bright is higher). The current mean decision boundary of the model is shown as a black line. Best viewed in color.

an expressive feature representation. For example, Sener and Savarese [5] only consider ConvNet representations learned on highly structured image data. In contrast, our work is inspired by Bayesian coresets [9, 10], which enable scalable Bayesian inference by approximating the log-likelihood of a labeled dataset with a sparse weighted subset thereof. Consequently, our method is less reliant on a structured feature space and only requires to evaluate log-likelihood terms.

## 7 Experiments and results

We perform experiments[3] to answer the following questions: (1) does our approach avoid correlated queries, (2) is our method competitive with greedy methods in the small-data regime, and (3) does our method scale to large datasets and models? We address questions (1) and (2) on several linear and probit regression tasks using the closed-form solutions derived in Section 4, and question (3) on large-scale regression and classification datasets by leveraging the projections from Section 5. Finally, we provide a runtime evaluation for all regression experiments. Full experimental details are deferred to Appendix C.

**Does our approach avoid correlated queries?** In Fig. 1, we have seen that traditional AL methods are prone to correlated queries. To investigate this further, in Fig. 2 we compare batches selected by ACS-FW and BALD on a simple probit regression task. Since BALD has no explicit batch construction mechanism, we naively choose the $b = 10$ most informative points according to BALD. While the BALD acquisition function does not change during batch construction, $\alpha_{\mathrm{ACS}}(\boldsymbol{x}_n; \mathcal{D}_0)$ rotates after each selected data point. This provides further intuition about why ACS-FW is able to spread the batch in data space, avoiding the strongly correlated queries that BALD produces.

**Is our method competitive with greedy methods in the small-data regime?** We evaluate the performance of ACS-FW on several UCI regression datasets. We compare against (i) RANDOM: select points randomly; (ii) MAXENT: naively construct batch using top $b$ points according to maximum entropy criterion (equivalent to BALD in this case); (iii) MAXENT-SG: use MAXENT with sequential greedy strategy (i.e. $b = 1$); (iv) MAXENT-I: sequentially acquire single data point, impute missing label and update model accordingly. Starting with 20 labeled points sampled randomly from the pool set, we use each AL method to iteratively grow the training dataset by requesting batches of size $b = 10$ until the budget of 100 queries is exhausted. To guarantee fair comparisons, all methods use the same neural linear model, i.e. a Bayesian linear regression model with a deterministic neural network feature extractor [19]. In this setting, posterior inference can be

Table 1: Final test RMSE on UCI regression datasets averaged over 40 (*year*: 5) seeds. MAXENT-I and MAXENT-SG require order(s) of magnitudes more model updates and are thus not directly comparable.

| | N | d | RANDOM | MAXENT | ACS-FW | MAXENT-I | MAXENT-SG |
|---|---|---|---|---|---|---|---|
| yacht | 308 | 6 | 1.272±0.0593 | **0.923±0.0319** | 1.031±0.0438 | 0.865±0.0276 | 0.971±0.0350 |
| boston | 506 | 13 | 4.068±0.0852 | **3.640±0.0652** | 3.799±0.0858 | 3.467±0.0676 | 3.458±0.0682 |
| energy | 768 | 8 | 0.959±0.0337 | 1.443±0.0857 | **0.855±0.0259** | 0.927±0.0461 | 1.055±0.0740 |
| power | 9568 | 4 | 5.108±0.0468 | 5.022±0.0428 | **4.984±0.0366** | 4.834±0.0313 | 4.855±0.0339 |
| year | 515 345 | 90 | 13.165±0.0307 | 13.030±0.0975 | **12.194±0.0596** | N/A | N/A |

Table 2: Runtime in seconds on UCI regression datasets averaged over 40 (*year*: 5) seeds. We report mean batch construction time (BT/it.) and total time (TT/it.) per AL iteration, as well as total cumulative time (total). MAXENT-I requires order(s) of magnitudes more model updates and is thus not directly comparable.

| | RANDOM | | | MAXENT | | | ACS-FW | | | MAXENT-I | | |
|---|---|---|---|---|---|---|---|---|---|---|---|---|
| | BT/it. | TT/it. | total | BT/it. | TT/it. | total | BT/it. | TT/it. | total | BT/it. | TT/it. | total |
| yacht | 0.0 | 8.9 | 88.6 | 1.3 | 10.2 | 101.7 | 0.0 | 9.1 | 107.2 | 12.3 | 105.7 | 1057.4 |
| boston | 0.0 | 12.4 | 123.6 | 2.4 | 14.5 | 144.8 | 0.1 | 12.4 | 132.7 | 23.5 | 157.9 | 1578.6 |
| energy | 0.0 | 12.1 | 121.4 | 3.9 | 16.0 | 159.6 | 0.1 | 12.6 | 137.8 | 37.5 | 170.5 | 1704.9 |
| power | 0.4 | 9.4 | 94.0 | 53.0 | 61.7 | 617.0 | 0.8 | 10.2 | 179.8 | 517.3 | 609.1 | 6090.7 |
| year | 30.2 | 381.2 | 3811.6 | 3391.5 | 3746.5 | 37 464.6 | 53.0 | 463.8 | 28 475.2 | N/A | N/A | N/A |

done in closed form [19]. The model is re-trained for 1000 epochs after every AL iteration using Adam [36]. After each iteration, we evaluate RMSE on a held-out set. Experiments are repeated for 40 seeds, using randomized 80/20% train-test splits. We also include a medium-scale experiment on *power* that follows the same protocol; however, for ACS-FW we use projections instead of the closed-form solutions as they yield improved performance and are faster. Further details, including architectures and learning rates, are in Appendix C.

The results are summarized in Table 1. ACS-FW consistently outperforms RANDOM by a large margin (unlike MAXENT), and is mostly on par with MAXENT on smaller datasets. While the results are encouraging, greedy methods such as MAXENT-SG and MAXENT-I still often yield better results in these small-data regimes. We conjecture that this is because single data points *do have* significant impact on the posterior. The benefits of using ACS-FW become clearer with increasing dataset size: as shown in Fig. 3, ACS-FW achieves much more data-efficient learning on larger datasets.

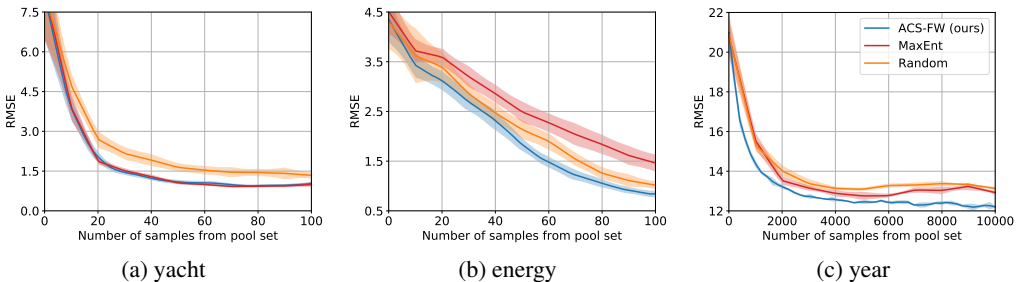

| (a) yacht | (b) energy | (c) year |

Figure 3: Test RMSE on UCI regression datasets averaged over 40 (a-b) and 5 (c) seeds during AL. Error bars denote two standard errors.

**Does our method scale to large datasets and models?**  Leveraging the projections from Section 5, we apply ACS-FW to large-scale datasets and complex models. We demonstrate the benefits of our approach on *year*, a UCI regression dataset with ca. 515 000 data points, and on the classification datasets *cifar10*, *SVHN* and *Fashion MNIST*. Methods requiring model updates after every data point (e.g. MAXENT-SG, MAXENT-I) are impractical in these settings due to their excessive runtime.

For *year*, we again use a neural linear model, start with 200 labeled points and allow for batches of size $b = 1000$ until the budget of 10 000 queries is exhausted. We average the results over 5 seeds,

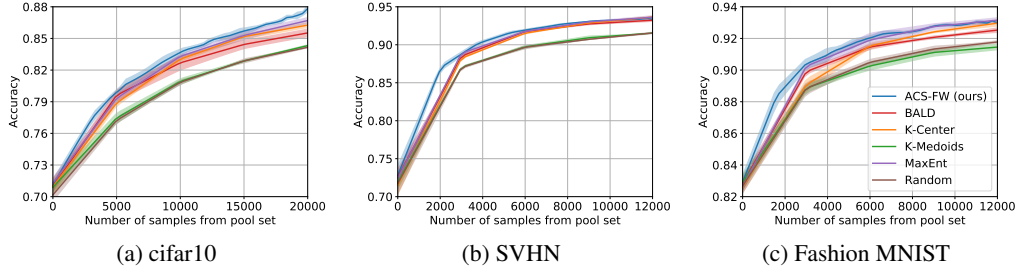

|                | (a) cifar10 | (b) SVHN | (c) Fashion MNIST |
| -------------- | ----------- | -------- | ----------------- |

Figure 4: Test accuracy on classification tasks over 5 seeds. Error bars denote two standard errors.

using randomized $80/20\%$ train-test splits. As can be seen in Fig. 3c, our approach significantly outperforms both RANDOM and MAXENT during the entire AL process.

For the classification experiments, we start with $1000$ (*cifar10*: $5000$) labeled points and request batches of size $b = 3000$ ($5000$), up to a budget of $12\,000$ ($20\,000$) points. We compare to RANDOM, MAXENT and BALD, as well as two batch AL algorithms, namely K-MEDOIDS and K-CENTER [5]. Performance is measured in terms of accuracy on a holdout test set comprising $10\,000$ (*Fashion MNIST*: $26\,032$, as is standard) points, with the remainder used for training. We use a neural linear model with a ResNet18 [16] feature extractor, trained from scratch at every AL iteration for $250$ epochs using Adam [36]. Since posterior inference is intractable in the multi-class setting, we resort to variational inference with mean-field Gaussian approximations [37, 38].

Fig. 4 demonstrates that in all cases ACS-FW significantly outperforms RANDOM, which is a strong baseline in AL [5, 13, 24]. Somewhat surprisingly, we find that the probabilistic methods (BALD and MAXENT), provide strong baselines as well, and consistently outperform RANDOM. We discuss this point and provide further experimental results in Appendix D. Finally, Fig. 4 demonstrates that in all cases ACS-FW performs at least as well as its competitors, including state-of-the-art non-probabilistic batch AL approaches such as K-CENTER. These results demonstrate that ACS-FW can usefully apply probabilistic reasoning to AL at scale, without any sacrifice in performance.

**Runtime Evaluation**    Runtime comparisons between different AL methods on the UCI regression datasets are shown in Table 2. For methods with fixed AL batch size $b$ (RANDOM, MAXENT and MAXENT-I), the number of AL iterations is given by the total budget divided by $b$ (e.g. $100/10 = 10$ for *yacht*). Thus, the total cumulative time (total) is given by the total time per AL iteration (TT/it.) times the number of iterations. MAXENT-I iteratively constructs the batch by selecting a single data point, imputing its label, and updating the model; therefore the batch construction time (BT/it.) and the total time per AL iteration take roughly $b$ times as long as for MAXENT (e.g. 10x for *yacht*). This approach becomes infeasible for very large batch sizes (e.g. 1000 for *year*). The same holds true for MAXENT-SG, which we have omitted here as the runtimes are similar to MAXENT-I. ACS-FW constructs batches of variable size, and hence the number of iterations varies.

As shown in Table 2, the batch construction times of ACS-FW are negligble compared to the total training times per AL iteration. Although ACS-FW requires more AL iterations than the other methods, the total cumulative runtimes are on par with MAXENT. Note that both MAXENT and MAXENT-I require to compute the entropy of a Student's T distribution, for which no batch version was available in PyTorch as we performed the experiments. Parallelizing this computation would likely further speed up the batch construction process.

## 8    Conclusion and future work

We have introduced a novel Bayesian batch AL approach based on sparse subset approximations. Our methodology yields intuitive closed-form solutions, revealing its connection to BALD as well as leverage scores. Yet more importantly, our approach admits relaxations (i.e. random projections) that allow it to tackle challenging large-scale AL problems with general non-linear probabilistic models. Leveraging the Frank-Wolfe weights in a principled way and investigating how this method interacts with alternative approximate inference procedures are interesting avenues for future work.

## Acknowledgments

Robert Pinsler receives funding from iCASE grant #1950384 with support from Nokia. Jonathan Gordon, Eric Nalisnick and José Miguel Hernández-Lobato were funded by Samsung Research, Samsung Electronics Co., Seoul, Republic of Korea. We thank Adrià Garriga-Alonso, James Requeima, Marton Havasi, Carl Edward Rasmussen and Trevor Campbell for helpful feedback and discussions.

## Footnotes

[1]Note that the entropy term in $\mathcal{L}_m$ (see Eq. (4)) vanishes under this norm as the gradient for $\boldsymbol{\theta}$ is zero.

[2] We only introduce $\alpha_{\text{ACS}}$ to compare to other acquisition functions; in practice we use Algorithm A.1.

[3]Source code is available at `https://github.com/rpinsler/active-bayesian-coresets`.

[4]We use $\alpha_{\mathrm{ACS}}$ (see Eq. (15)) as an acquisition function for ACS-FW only for the sake of visualization.

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
