[Supplementary Material · Supplementary_Bayesian_Batch_Active_Learning_as_Sparse_Subset_Approximation.pdf]

# Supplementary Material for Bayesian Batch Active Learning as Sparse Subset Approximation

**Robert Pinsler**
Department of Engineering
University of Cambridge
rp586@cam.ac.uk

**Jonathan Gordon**
Department of Engineering
University of Cambridge
jg801@cam.ac.uk

**Eric Nalisnick**
Department of Engineering
University of Cambridge
etn22@cam.ac.uk

**José Miguel Hernández-Lobato**
Department of Engineering
University of Cambridge
jmh233@cam.ac.uk

## A    Algorithms

### A.1    Active Bayesian coresets with Frank-Wolfe optimization (ACS-FW)

Algorithm A.1 outlines the ACS-FW procedure for a budget $b$, vectors $\{\mathcal{L}_n\}_{n=1}^N$ and the choice of an inner product $< \cdot, \cdot >$ (see Section 2). After computing the norms $\sigma_n$ and $\sigma$ (Lines 2 and 3) and initializing the weight vector $\boldsymbol{w}$ to zero (Line 4), the algorithm performs $b$ iterations of Frank-Wolfe optimization. At each iteration, the Frank-Wolfe algorithm chooses exactly one data point (which can be viewed as nodes on the polytope) to be added to the batch (Line 6). The weight update for this data point can then be computed by performing a line search in closed form [1] (Line 7), and using the step-size to update $\boldsymbol{w}$ (Line 8). Finally, the optimal weight vector with cardinality $\leq b$ is returned. In practice, we project the weights back to the feasible space by binarizing them (not shown; see Section 2 for more details), as working with the continuous weights directly is non-trivial.

### A.2    ACS-FW with random projections

Algorithm A.2 details the process of constructing an AL batch with budget $b$ and $J$ random feature projections for the weighted Euclidean inner product from Eq. (16).

## B    Closed-form derivations

### B.1    Linear regression

Consider the following model for scalar Bayesian linear regression,

$$y_n = \boldsymbol{\theta}^T \boldsymbol{x}_n + \epsilon_n, \quad \epsilon_n \sim \mathcal{N}(0, \sigma_0^2), \quad \boldsymbol{\theta} \sim p(\boldsymbol{\theta}),$$

where $p(\boldsymbol{\theta})$ denotes the prior. To avoid notational clutter we assume a factorized Gaussian prior with unit variance, but what follows is easily extended to richer Gaussian priors. Given an initial labeled dataset $\mathcal{D}_0$, the parameter posterior can be computed in closed form as

$$p(\boldsymbol{\theta}|\mathcal{D}_0, \sigma_0^2) = \mathcal{N}(\boldsymbol{\theta}; \boldsymbol{\mu_\theta}, \boldsymbol{\Sigma_\theta}) \tag{B.18}$$
$$\boldsymbol{\mu_\theta} = \left(\boldsymbol{X}^T \boldsymbol{X} + \sigma_0^2 \boldsymbol{I}\right)^{-1} \boldsymbol{X}^T \boldsymbol{y}$$
$$\boldsymbol{\Sigma_\theta} = \sigma_0^2 \left(\boldsymbol{X}^T \boldsymbol{X} + \sigma_0^2 \boldsymbol{I}\right)^{-1},$$

---
**Algorithm A.1** Active Bayesian Coresets with Frank-Wolfe Optimization
---
1: **procedure** ACS-FW$(b, \{\mathcal{L}_n\}_{n=1}^N, < \cdot, \cdot >)$
2:     $\sigma_n \leftarrow \sqrt{\langle \mathcal{L}_n, \mathcal{L}_n \rangle} \quad \forall n$                                                                   ▷ Compute norms
3:     $\sigma \leftarrow \sum_n \sigma_n$
4:     $\boldsymbol{w} \leftarrow \boldsymbol{0}$                                                                                                  ▷ Initialize weights to 0
5:     **for** $t \in 1, ..., b$ **do**
6:         $f \leftarrow \underset{n \in N}{\arg \max} \left[ \left\langle \mathcal{L} - \mathcal{L}(\boldsymbol{w}), \frac{1}{\sigma_n} \mathcal{L}_n \right\rangle \right]$                         ▷ Greedily select point $f$
7:         $\gamma \leftarrow \dfrac{\left[ \left\langle \frac{\sigma}{\sigma_f} \mathcal{L}_f - \mathcal{L}(\boldsymbol{w}), \mathcal{L} - \mathcal{L}(\boldsymbol{w}) \right\rangle \right]}{\left[ \left\langle \frac{\sigma}{\sigma_f} \mathcal{L}_f - \mathcal{L}(\boldsymbol{w}), \frac{\sigma}{\sigma_f} \mathcal{L}_f - \mathcal{L}(\boldsymbol{w}) \right\rangle \right]}$               ▷ Perform line search for step-size $\gamma$
8:         $\boldsymbol{w} \leftarrow (1 - \gamma)\boldsymbol{w} + \gamma \frac{\sigma}{\sigma_f} \boldsymbol{1}_f$                                     ▷ Update weight for newly selected point
9:     **end for**
10:     **return** $w$
11: **end procedure**
---

---
**Algorithm A.2** ACS-FW with Random Projections (for Weighted Euclidean Inner Product)
---
1: **procedure** ACS-FW$(b, J)$
2:     $\boldsymbol{\theta}_j \sim \hat{\pi} \quad j = 1, \ldots, J$                                                       ▷ Sample parameters
3:     $\hat{\mathcal{L}}_n = \frac{1}{\sqrt{J}} \left[ \mathcal{L}_n(\boldsymbol{\theta}_1), \cdots, \mathcal{L}_n(\boldsymbol{\theta}_J) \right]^T \quad \forall n$          ▷ Compute random feature projections
4:     **return** ACS-FW$(b, \{\hat{\mathcal{L}}_n\}_{n=1}^N, (\cdot)^T(\cdot))$                       ▷ Call Algorithm A.1 using projections
5: **end procedure**
---

and the predictive posterior is given by

$$
\begin{aligned}
p(y_n | \boldsymbol{x}_n, \mathcal{D}_0, \sigma_0^2) &= \int_{\boldsymbol{\theta}} p(y_n | \boldsymbol{x}_n, \boldsymbol{\theta}) p(\boldsymbol{\theta} | \mathcal{D}_0, \sigma_0^2) \, d\boldsymbol{\theta} \\
&= \mathcal{N}(y_n; \boldsymbol{\mu}_{\boldsymbol{\theta}}^T \boldsymbol{x}_n, \sigma_0^2 + \boldsymbol{x}_n^T \boldsymbol{\Sigma}_{\boldsymbol{\theta}} \boldsymbol{x}_n).
\end{aligned}
\tag{B.19}
$$

Using this model, we can derive a closed-form term for the inner product in Eq. (8),

$$
\begin{aligned}
\langle \mathcal{L}_n, \mathcal{L}_m \rangle_{\hat{\pi}, \mathcal{F}} &= \mathbb{E}_{\hat{\pi}} \left[ (\nabla_{\boldsymbol{\theta}} \mathcal{L}_n)^T (\nabla_{\boldsymbol{\theta}} \mathcal{L}_m) \right] \\
&= \mathbb{E}_{\hat{\pi}} \left[ \left( \frac{1}{\sigma_0^2} (\mathbb{E}[y_n] - \boldsymbol{x}_n^T \boldsymbol{\theta}) \boldsymbol{x}_n \right)^T \left( \frac{1}{\sigma_0^2} (\mathbb{E}[y_m] - \boldsymbol{x}_m^T \boldsymbol{\theta}) \boldsymbol{x}_m \right) \right] \\
&= \frac{\boldsymbol{x}_n^T \boldsymbol{x}_m}{\sigma_0^4} \mathbb{E}_{\hat{\pi}} \left[ \left( \boldsymbol{\mu}_{\boldsymbol{\theta}}^T \boldsymbol{x}_n - \boldsymbol{\theta}^T \boldsymbol{x}_n \right)^T \left( \boldsymbol{\mu}_{\boldsymbol{\theta}}^T \boldsymbol{x}_m - \boldsymbol{\theta}^T \boldsymbol{x}_m \right) \right] \\
&= \frac{\boldsymbol{x}_n^T \boldsymbol{x}_m}{\sigma_0^4} \left( \boldsymbol{x}_n^T \boldsymbol{\Sigma}_{\boldsymbol{\theta}} \boldsymbol{x}_m \right),
\end{aligned}
$$

where in the second equality we have taken expectation w.r.t. $p(y_n | \boldsymbol{x}_n, \mathcal{D}_0, \sigma_0^2)$ from Eq. (B.19), and in the third equality w.r.t. $\hat{\pi} = p(\boldsymbol{\theta} | \mathcal{D}_0, \sigma_0^2)$ from Eq. (B.18). Similarly, we obtain

$$
\langle \mathcal{L}_n, \mathcal{L}_n \rangle_{\hat{\pi}, \mathcal{F}} = \frac{\boldsymbol{x}_n^T \boldsymbol{x}_n}{\sigma_0^4} \left( \boldsymbol{x}_n^T \boldsymbol{\Sigma}_{\boldsymbol{\theta}} \boldsymbol{x}_n \right).
$$

For this model, BALD [2, 3] can also be evaluated in closed form:

$$
\begin{aligned}
\alpha_{\text{BALD}}(\boldsymbol{x}_n; \mathcal{D}_0) &= \mathbb{H} \left[ \boldsymbol{\theta} | \mathcal{D}_0, \sigma_0^2 \right] - \mathbb{E}_{p(y_n | \boldsymbol{x}_n, \mathcal{D}_0)} \left[ \mathbb{H} \left[ \boldsymbol{\theta} | \boldsymbol{x}_n, y_n, \mathcal{D}_0, \sigma_0^2 \right] \right] \\
&= \frac{1}{2} \mathbb{E}_{\hat{\pi}} \left[ \log \frac{\sigma_0^2 + \boldsymbol{x}_n^T \boldsymbol{\Sigma}_{\boldsymbol{\theta}} \boldsymbol{x}_n}{\sigma_0^2} + \frac{\sigma_0^2 + (\boldsymbol{\mu}_{\boldsymbol{\theta}}^T \boldsymbol{x}_n - \boldsymbol{\theta}^T \boldsymbol{x}_n)^2}{\sigma_0^2 + \boldsymbol{x}_n^T \boldsymbol{\Sigma}_{\boldsymbol{\theta}} \boldsymbol{x}_n} - 1 \right] \\
&= \frac{1}{2} \log \left( \sigma_0^2 + \frac{\boldsymbol{x}_n^T \boldsymbol{\Sigma}_{\boldsymbol{\theta}} \boldsymbol{x}_n}{\sigma_0^2} \right).
\end{aligned}
$$

(a) $\alpha_{\text{BALD}}$          (b) $\alpha_{\text{ACS}}/\boldsymbol{x}_n^T\boldsymbol{x}_n$

Figure B.5: $\alpha_{\text{BALD}}$ and $\alpha_{\text{ACS}}$ (without the magnitude term) evaluated on synthetic data drawn from a linear regression model with $y_n = x_n + \epsilon$, where $\epsilon \sim \mathcal{N}(0,5)$. $\alpha_{\text{BALD}}$ and $\alpha_{\text{ACS}}/\boldsymbol{x}_n^T\boldsymbol{x}_n$ are equivalent (up to a constant factor) in this model.

We can make a direct comparison with BALD by treating the squared norm of a data point with itself as an acquisition function, $\alpha_{\text{ACS}}(\boldsymbol{x}_n; \mathcal{D}_0) = \langle \mathcal{L}_n, \mathcal{L}_n \rangle_{\hat{\pi}, \mathcal{F}}$, yielding,

$$\alpha_{\text{ACS}}(\boldsymbol{x}_n; \mathcal{D}_0) = \frac{\boldsymbol{x}_n^T\boldsymbol{x}_n}{\sigma_0^4}\boldsymbol{x}_n^T\boldsymbol{\Sigma}_{\boldsymbol{\theta}}\boldsymbol{x}_n.$$

Viewing $\alpha_{\text{ACS}}$ as a greedy acquisition function is reasonable as (i) the norm of $\mathcal{L}_n$ is related to the magnitude of the reduction in Eq. (5), and thus can be viewed as a proxy for greedy optimization. (ii) This establishes a link to notions of *sensitivity* from the original work on Bayesian coresets [1, 4], where $\sigma_n = \|\mathcal{L}_n\|$ is the key quantity for constructing the coreset (i.e. by using it for importance sampling or Frank-Wolfe optimization).

As demonstrated in Fig. B.5, dropping $\boldsymbol{x}_n^T\boldsymbol{x}_n$ from $\alpha_{\text{ACS}}$ makes the two quantities proportional— $\exp(2\alpha_{\text{BALD}}(\boldsymbol{x}_n; \mathcal{D}_0)) \propto \alpha_{\text{ACS}}(\boldsymbol{x}_n; \mathcal{D}_0)$—and thus equivalent under a greedy maximizer.

## B.2 Logistic regression and probit regression

The probit regression model used in the main section of the paper is closely related to logistic regression. Since the latter is more common in pratice, we will start from a Bayesian logistic regression model and apply the standard probit approximation to render inference tractable.

Consider the following Bayesian logistic regression model,

$$p(y_n|\boldsymbol{x}_n, \boldsymbol{\theta}) = \text{Ber}\left(\sigma(\boldsymbol{\theta}^T\boldsymbol{x}_n)\right), \quad \sigma(z) := \frac{1}{1 + \exp(-z)}, \quad \boldsymbol{\theta} \sim p(\boldsymbol{\theta}),$$

where we again assume $p(\boldsymbol{\theta})$ is a factorized Gaussian with unit variance. The exact parameter posterior distribution is intractable for this model due to the non-linear likelihood. We assume an approximation of the form $p(\boldsymbol{\theta}|\mathcal{D}_0) \approx \mathcal{N}(\boldsymbol{\theta}; \boldsymbol{\mu}_{\boldsymbol{\theta}}, \boldsymbol{\Sigma}_{\boldsymbol{\theta}})$. More importantly, the posterior predictive is also intractable in this setting. For the purpose of this derivation, we use the additional approximation

$$p(y_n|\boldsymbol{x}_n, \mathcal{D}_0) = \int_{\boldsymbol{\theta}} p(y_n|\boldsymbol{x}_n, \boldsymbol{\theta})p(\boldsymbol{\theta}|\mathcal{D}_0)\,d\boldsymbol{\theta}$$

$$\approx \int_{\boldsymbol{\theta}} \Phi(\boldsymbol{\theta}^T\boldsymbol{x}_n)\mathcal{N}(\boldsymbol{\theta}; \boldsymbol{\mu}_{\boldsymbol{\theta}}, \boldsymbol{\Sigma}_{\boldsymbol{\theta}})\,d\boldsymbol{\theta}$$

$$= \text{Ber}\left(\Phi\left(\frac{\boldsymbol{\mu}_{\boldsymbol{\theta}}^T\boldsymbol{x}_n}{\sqrt{1 + \boldsymbol{x}_n^T\boldsymbol{\Sigma}\boldsymbol{x}_n}}\right)\right),$$

where in the second line we have plugged in our approximation to the parameter posterior, and used the well-known approximation $\sigma(z) \approx \Phi(z)$, where $\Phi(\cdot)$ represents the standard Normal cdf [5].

Next, we derive a closed-form approximation for the weighted Fisher inner product in Eq. (8). We begin by noting that

$$\langle \mathcal{L}_n, \mathcal{L}_m \rangle_{\hat{\pi}, \mathcal{F}} \approx \boldsymbol{x}_n^T \boldsymbol{x}_m \left( \underset{\hat{\pi}}{\mathbb{E}} \left[ \Phi\left(\boldsymbol{\theta}^T \boldsymbol{x}_n\right) \Phi\left(\boldsymbol{\theta}^T \boldsymbol{x}_m\right) \right] - \Phi(\zeta_n)\Phi(\zeta_m) \right), \qquad \text{(B.20)}$$

where we define $\zeta_i = \frac{\boldsymbol{\mu}_{\boldsymbol{\theta}}^T \boldsymbol{x}_i}{\sqrt{1 + \boldsymbol{x}_i^T \boldsymbol{\Sigma}_{\boldsymbol{\theta}} \boldsymbol{x}_i}}$, and use $\sigma(z) \approx \Phi(z)$ as before. Next, we employ the identity [6]

$$\int \Phi(a + bz)\Phi(c + dz)\mathcal{N}(z; 0, 1)dz = \text{BvN}\left( \frac{a}{\sqrt{1 + b^2}}, \frac{c}{\sqrt{1 + d^2}}, \rho = \frac{bd}{\sqrt{1 + b^2}\sqrt{1 + d^2}} \right),$$

where $\text{BvN}(a, b, \rho)$ is the bi-variate Normal (with correlation $\rho$) cdf evaluated at $(a, b)$. Plugging this, and Eq. (B.22) into Eq. (B.20) yields

$$\underset{\hat{\pi}}{\mathbb{E}} \left[ \left(\nabla_{\boldsymbol{\theta}} \mathcal{L}_n\right)^T \left(\nabla_{\boldsymbol{\theta}} \mathcal{L}_m\right) \right] \approx \boldsymbol{x}_n^T \boldsymbol{x}_m \left( \text{BvN}\left(\zeta_n, \zeta_m, \rho_{n,m}\right) - \Phi(\zeta_m)\Phi(\zeta_m) \right),$$

where $\rho_{n,m} = \frac{\boldsymbol{x}_n^T \boldsymbol{\Sigma}_{\boldsymbol{\theta}} \boldsymbol{x}_m}{\sqrt{1 + \boldsymbol{x}_n^T \boldsymbol{\Sigma}_{\boldsymbol{\theta}} \boldsymbol{x}_n}\sqrt{1 + \boldsymbol{x}_n^T \boldsymbol{\Sigma}_{\boldsymbol{\theta}} \boldsymbol{x}_m}}$.

Next, we derive an expression for the squared norm, i.e.

$$\begin{aligned} \langle \mathcal{L}_n, \mathcal{L}_n \rangle_{\hat{\pi}, \mathcal{F}} &= \underset{\hat{\pi}}{\mathbb{E}} \left[ \left(\nabla_{\boldsymbol{\theta}} \mathcal{L}_n\right)^T \left(\nabla_{\boldsymbol{\theta}} \mathcal{L}_n\right) \right] \\ &= \underset{\hat{\pi}}{\mathbb{E}} \left[ \left(\left(\mathbb{E}[y_n] - \sigma\left(\boldsymbol{x}_n^T \boldsymbol{\theta}\right)\right) \boldsymbol{x}_n\right)^T \left(\left(\mathbb{E}[y_n] - \sigma\left(\boldsymbol{x}_n^T \boldsymbol{\theta}\right)\right) \boldsymbol{x}_n\right) \right] \\ &= \boldsymbol{x}_n^T \boldsymbol{x}_n \left( \Phi(\zeta_n)^2 - 2\Phi(\zeta_n) \underset{\hat{\pi}}{\mathbb{E}} \left[ \sigma\left(\boldsymbol{\theta}^T \boldsymbol{x}_n\right) \right] + \underset{\hat{\pi}}{\mathbb{E}} \left[ \sigma\left(\boldsymbol{\theta}^T \boldsymbol{x}_n\right)^2 \right] \right). \end{aligned} \qquad \text{(B.21)}$$

Here, we again use the approximation $\sigma(z) \approx \Phi(z)$, and the following identity [6]:

$$\int \left(\Phi\left(\boldsymbol{\theta}^T \boldsymbol{x}\right)\right)^2 \mathcal{N}\left(\boldsymbol{\theta}; \boldsymbol{\mu}_{\boldsymbol{\theta}}, \boldsymbol{\Sigma}_{\boldsymbol{\theta}}\right) \mathrm{d}\boldsymbol{\theta} = \Phi\left(\zeta\right) - 2\text{T}\left( \zeta, \frac{1}{\sqrt{1 + 2\boldsymbol{x}^T \boldsymbol{\Sigma}_{\boldsymbol{\theta}} \boldsymbol{x}}} \right), \qquad \text{(B.22)}$$

where $\text{T}(\cdot, \cdot)$ is Owen's T function[5] [6]. Plugging Eq. (B.22) back into Eq. (B.21) and taking expectation w.r.t. the approximate posterior, we have that

$$\underset{\hat{\pi}}{\mathbb{E}} \left[ \left(\nabla_{\boldsymbol{\theta}} \mathcal{L}_n\right)^T \left(\nabla_{\boldsymbol{\theta}} \mathcal{L}_n\right) \right] = \boldsymbol{x}_n^T \boldsymbol{x}_n \left( \Phi\left(\zeta_n\right)\left(1 - \Phi\left(\zeta_n\right)\right) - 2\text{T}\left( \zeta_n, \frac{1}{\sqrt{1 + 2\boldsymbol{x}_n^T \boldsymbol{\Sigma}_{\boldsymbol{\theta}} \boldsymbol{x}_n}} \right) \right).$$

## C   Experimental details

**Computing infrastructure**   All experiments were run on a desktop Ubuntu 16.04 machine. We used an Intel Core i7-3820 @ 3.60GHz x 8 CPU for experiments on *yacht*, *boston*, *energy* and *power*, and a GeForce GTX TITAN X GPU for all others.

**Hyperparameter selection**   We manually tuned the hyper-parameters with the goal of trading off performance and stability of the model training throughout the AL process, while keeping the protocol similar across datasets. Although a more systematic hyper-parameter search might yield improved results, we anticipate that the gains would be comparable across AL methods since they all share the same model and optimization procedure.

### C.1   Regression experiments

**Model**   We use a deterministic feature extractor consisting of two fully connected hidden layers with 30 (*year*: 100) units, interspersed with batch norm and ReLU activation functions. Weights

and biases are initialized from $\mathcal{U}(-\sqrt{k}, \sqrt{k})$, where $k = 1/N_{\text{in}}$, and $N_{\text{in}}$ is the number of incoming features. We additionally apply L2 weight decay with regularization parameter $\lambda = 1$ (*power*, *year*: $\lambda = 3$). The final layer performs exact Bayesian inference. We place a factorized zero-mean Gaussian prior with unit variance on the weights of the last layer $L$, $\boldsymbol{\theta}_L \sim \mathcal{N}(\boldsymbol{\theta}_L; \mathbf{0}, \boldsymbol{I})$, and an inverse Gamma prior on the noise variance, $\sigma_0^2 \sim \Gamma^{-1}(\sigma_0^2; \alpha_0, \beta_0)$, with $\alpha_0 = 1, \beta_0 = 1$ (*power*, *year*: $\beta_0 = 3$). Inference with this prior can be performed in closed form, where the predictive posterior follows a Student's T distribution [7]. For *power* and *year*, we use $J = 10$ projections during the batch construction of ACS-FW.

**Optimization**  Inputs and outputs are normalized during training to have zero mean and unit variance, and un-normalized for prediction. The network is trained for $1000$ epochs with the Adam optimizer, using a learning rate of $\alpha = 10^{-2}$ (*power*, *year*: $10^{-3}$) and cosine annealing. The training batch size is adapted during the AL process as more data points are acquired: we set the batch size to the closest power of $2 \le |\mathcal{D}_0|/2$ (e.g. for *boston* we initially start with a batch size of 8), but not more than $512$. For *power* and *yacht*, we divert from this protocol to stabilize the training process, and set the batch size to $\min(|\mathcal{D}_0|, 32)$.

### C.2  Classification experiments

**Model**  We employ a deterministic feature extractor consisting of a ResNet-18 [8], followed by one fully-connected hidden layer with 32 units with a ReLU activation function. All weights are initialized with Glorot initialization [9]. We additionally apply L2 weight decay with regularization parameter $\lambda = 5 \cdot 10^{-4}$ to all weights of this feature extractor. The final layer is a dense layer that returns samples using local reparametrization [10], followed by a softmax activation function. The mean weights of the last layer are initialized from $\mathcal{N}(0, 0.05)$ and the log standard deviation weights of the variances are initialized from $\mathcal{N}(-4, 0.05)$. We place a factorized zero-mean Gaussian prior with unit variance on the weights of the last layer $L$, $\boldsymbol{\theta}_L \sim \mathcal{N}(\boldsymbol{\theta}_L; \mathbf{0}, \boldsymbol{I})$. Since exact inference is intractable, we perform mean-field variational inference [11, 12] on the last layer. The predictive posterior is approximated using 100 samples. We use $J = 10$ projections during the batch construction of ACS-FW.

**Optimization**  We use data augmentation techniques during training, consisting of random cropping to 32px with padding of 4px, random horizontal flipping and input normalization. The entire network is trained jointly for $1000$ epochs with the Adam optimizer, using a learning rate of $\alpha = 10^{-3}$, cosine annealing, and a fixed training batch size of 256.

## D  Probabilistic methods for active learning

One surprising result we found in our experiments was the strong performance of the probabilistic baselines MAXENT and BALD, especially considering that a number of previous works have reported weaker results for these methods (e.g. [13]).

Probabilistic methods rely on the parameter posterior distribution $p(\boldsymbol{\theta}|\mathcal{D}_0)$. For neural network based models, posterior inference is usually intractable and we are forced to resort to approximate inference techniques [14]. We hypothesize that probabilistic AL methods are highly sensitive to the inference method used to train the approximate posterior distribution $q(\boldsymbol{\theta}) \approx p(\boldsymbol{\theta}|\mathcal{D}_0)$. Many works use Monte Carlo Dropout (MCDropout) [15] as the standard method for these approximations [16, 17], but commonly only use MCDropout on the final layer.

In our work, we find that a Bayesian multi-class classification model on the final layer of a powerful deterministic feature extractor, trained with variational inference [11, 12] tends to lead to significant performance gains compared to using MCDropout on the final layer. A comparison of these two methods is shown in Fig. D.6, demonstrating that for *cifar10*, *SVHN* and *Fashion MNIST* a neural linear model is preferable to one trained with MCDropout in the AL setting. In future work, we intend to further explore the trade-offs implied by using different inference procedures for AL.

| | | |
|---|---|---|
| (a) cifar10 | (b) SVHN | (c) Fashion MNIST |

Figure D.6: Test accuracy on classification tasks over 5 seeds. Error bars denote two standard errors.

## Footnotes

[5]Efficient open-source implementations of numerical approximations exist, e.g. in scipy.