[Reviews · NeurIPS 2019]

Reviewer 1



[review updated in Improvements section] The paper is very well written. The paper tackles the problem of sampling for Active learning such that a mini-batch of examples is diverse. It proposes a Bayesian approach as a solution. In order to resolve non-tractability of the original problem, the authors take expectation of outcomes w.r.t. the current predictive posterior distribution, and Bayesian core-sets (which calculates total expectation by constructing a sparse subset approximation). A tricky part of the approach in practive is to construct a suitable inner product, the authors suggest some variants for generalized linear models, and random projections for non-linear models. In the experiments on synthetic and real data from UCI, MNIST, cifar10, SVHN, and fashion MNIST, the authors demonstrate that the method achives what it was purposed to achieve (diversity, competitiveness, scalability). I think the paper is a good contribution. It solves an important problem, suggests a novel algorithm, and has thorough evaluations of the most important aspects of the algorith. I also noticed the authors haven't referenced one of the recent relevant works (I think it was on arxiv only), "Diverse mini-batch Active Learning", which might add to their baselines.

Reviewer 2



This manuscript proposes a novel method for Bayesian batch active learning through sparse subset approximation and a convenient set of reductions to arrive at a tractable algorithm. This method is validated and explored through a series of special cases (linear regression and classification), illustrations, and experiments. Overall the method appears to be competitive with the state of the art. Overall this manuscript is well written, insightful, and enjoyable to read. The proposed approach outlined on page 3 is elegant, appears to work well in practice, and the approach may be useful in other settings. I only have a few concerns about the work as presented that I will outline below. These are mostly in linear order: - The fact that naive adaptation of sequential active learning algorithms to the batch setting by ranking and taking the top-scoring points produces highly correlated batches is completely unsurprising. However, there is a simple mechanism that can be used to address this phenomenon, which has for example been used in Bayesian optimization [1][2]: - begin with empty batch - for i = 1 .. batch size - choose next point x using sequential AL - impute observation y for this point (e.g., by sampling or MAP) - add x to batch - update model given (x, y) - return batch I _think_ this may be the "greedy" procedure you mention in for final experiment but I'm not sure. Obviously there is a tradeoff here in that you must do (batch size) model updates to compute the batch, although the overall running time is only linearly more expensive than choosing a single point. By imputing observations you get a natural "repulsive effect" lessening the correlations among batch members. I think at the very least some more discussion is warranted on this point. Added after response: after some reflection, I am lowering my score slightly since this simple and pervasive batch construction strategy is completely ignored and note even acknowledged in the discussion. The paper would be much stronger (and more intellectually honest) if it were discussed and evaluated, even if the evaluation concluded the runtime was too great. I don't buy the idea that a linear slowdown in active learning (where evaluations are expensive) is as huge a deal as the authors suggest. - The black box in (5) should be a w. - I think the discussion on logistic regression would be more clear if you simply replaced it entirely by probit regression. Why mention the logistic function at all if you're just going to throw it away a few lines later? - I disagree that extending entropy-based methods to the batch setting is necessarily difficult. For example in logistic regression it would be trivial to at least greedily maximize the entropy (the determinant of the predictive covariance matrix) offline using a series of rank-1 updates. This is effectively the above procedure although in this particular case you don't need to impute y at all. - The manuscript would be strengthened with some mention/study of the running time of the proposed method compared to the others. [1]: https://hal.archives-ouvertes.fr/hal-00260579/document [2]: Jiang, et al. Efficient nonmyopic batch active search. NeurIPS 2018

Reviewer 3



The paper considers the problem of selecting an optimal batch of samples for evaluation to increase the predictive performance of the model over the entire dataset, aka active learning. The paper considers a Bayesian perspective, where the goal is to select a number of points (fixed budget) to minimize the difference between the resulting posterior and the posterior one could have had if all the points were labeled. The problem is formulated as a sparse data approximation problem and corresponding algorithms are presented. Some derivations and interpretations are made in simple models (linear, logistic). Also the algorithm is extended to deal with larger datasets through the use of random projections. A collection of experimental results for regression and classification on publicly available datasets are presented. Originality: As far as I know the algorithmic development, the interpretations and the experimental results are original. Quality: The material appears to be sound, although some derivations are presented in the supplementary material which I didn’t check. I would like to have a few things clarified: There has been other work on Bayesian active learning that seems to fit well the batch setting that has not mentioned: “Nonmyopic active learning of Gaussian processes: an exploration-exploitation approach” by Krause et al. Although the focus is on GP, I find the discussion and derivations very relevant to the general Bayesian setting. What is the relationship with this work? If this is not suitable for batch AL, why? If yes, how is it different? What is the relationship between the objective proposed in this paper and one of the objectives discussed there? Please provide a careful derivation of Eq.4, as in its current form it is not clear where the first equality is coming from Clarity: The paper is well written overall. However the motivation for the batch AL setting can be improved. Typically in AL the cost of a label is significantly more expensive than computation time. Yet in this case the argument is that the label cost is “cheaper” then updating a model, leading to grouping of samples. Indeed, what are the real applications for this scenario? Significance: Given concerns about the motivation I find this work of somewhat limited significance. Evaluations on the MNIST datasets, although useful, do not relieve those concerns. In other words, experiments on a dataset where labeling costs are “on par” with model update time would be ideal. Post rebuttal feedback: Thank you for clarifying some of the concerns I raised, including those in the paper will certainly improve it. That being said, the point raised by R2 about the simple way to construct the batch through imputation and model update is still valid. The paper would have been much better if either: experimental results comparing with this method are done; or discussion with sufficient details is provided regarding the cases where model update is too expensive to run. So far I can think of this issue possibly happening in the parallel simulation setting only, but none of the other motivational examples mentioned. As a result, my score remains the same.

[Author Response · NeurIPS 2019]

We thank the reviewers for their time and effort reviewing our paper. We are pleased that you found our work to "solve an important problem" (R1), our method to be "elegant" (R2), and our paper to be "well written" (R3). Below we respond to the key comments and issues in order of apperance.

**R1: "Diverse mini-batch Active Learning" and open-source code** We will discuss the paper and consider it as an additional baseline. At the latest, source code will be made publicly available upon publication.

**R2: Batch active learning by imputing labels** When we write "greedy," we are referring to the sequential setting in which we alternate between querying a single data point and updating the model. When we say "naive batch" we mean ranking and taking the top $b$ points. The proposed imputation approach could be considered as an alternative batch construction method not currently discussed in the paper. The key issue with this method is indeed the cost of performing a model update after each label is imputed. While a linear increase in runtime due to repeated model updates may be acceptable for constructing smaller-sized batches, in the large-scale settings we consider it simply becomes infeasible. Interestingly, there is a close relationship between the imputation approach and our method: in fact, our method also uses the expected (i.e., imputed) labels under the model to construct the batch in a principled way; however, importantly, it does not require making repeated predictions over the entire pool set or updating the model after each point is added to the batch. We will add a thorough discussion on this topic to a future version of the paper.

**R2: Logistic vs. probit regression** We agree with R2 that the discussion on logistic regression is clearer by considering probit regression to begin with. We will update the section accordingly.

**R2: Extending entropy-based methods to the batch setting is not necessarily difficult** We agree the wording is misleading. The main issue is how to do this efficiently for complex, non-linear models. We will clarify this point.

**R2: Runtime discussion** In general, constructing the batches has negligible cost (cf. Section 5 for a discussion on computational complexity) compared to updating the model, so increasing the batch size allows to decrease overall computational cost. We will add a more detailed comparison with other methods to the experimental section.

**R3: "Nonmyopic active learning of Gaussian processes: an exploration-exploitation approach"** While GPs suffer from scalability issues and cannot be applied to many of the domains we considered, we agree that batch active learning (AL) with GPs is under-discussed in the paper. Krause & Guestrin (2007) consider mutual information (MI; also known as BALD) as an acquisition criterion, which makes this work related to ours. However, they immediately reduce the batch formulation to the sequential greedy case. We highlight similarities and differences to sequential greedy MI/BALD in Sections 4 and 6, and will add a discussion on extending sequential greedy methods to the batch setting (cf. *R2: Extending entropy-based methods to the batch setting is not necessarily difficult*). Less relevant for our work are GP-specific details (as we focus on BNNs) and proofs relying on submodularity (as the diminishing returns property does not necessarily hold when performing approximate inference and stochastic optimization). We will expand the related work section accordingly.

**R3: Clarifying first equality in eq. (4)** The first equality is achieved by applying Bayes' rule to the posterior (eq. (1) in the main paper), taking the logarithm, and applying linearity of expectations:

$$\mathbb{E}_{\mathcal{Y}_p|\mathcal{X}_p,\mathcal{D}_0}[\log p(\boldsymbol{\theta}|\mathcal{D}_0 \cup (\mathcal{X}_p,\mathcal{Y}_p))] = \mathbb{E}_{\mathcal{Y}_p|\mathcal{X}_p,\mathcal{D}_0}[\log p(\boldsymbol{\theta}|\mathcal{D}_0) + \log p(\mathcal{Y}_p|\mathcal{X}_p,\boldsymbol{\theta}) - \log p(\mathcal{Y}_p|\mathcal{X}_p,\mathcal{D}_0)]$$
$$= \log p(\boldsymbol{\theta}|\mathcal{D}_0) + \mathbb{E}_{\mathcal{Y}_p|\mathcal{X}_p,\mathcal{D}_0}[\log p(\mathcal{Y}_p|\mathcal{X}_p,\boldsymbol{\theta})] + \mathbb{H}[\mathcal{Y}_p|\mathcal{X}_p,\mathcal{D}_0], \tag{1}$$

where we used $\mathbb{E}_{\mathcal{Y}_p}[-\log p(\mathcal{Y}_p|\mathcal{X}_p,\mathcal{D}_0)] = \mathbb{H}[\mathcal{Y}_p|\mathcal{X}_p,\mathcal{D}_0]$. We will make this derivation clearer in the next version.

**R3: Motivation for the batch AL setting can be improved** This scenario is practical in a number of real-world applications, particularly when the cost of acquiring labels is high but can be parallelized. Examples include crowd-sourcing a complex labeling task, leveraging parallel simulations on a compute cluster, or performing experiments that require resources with time-limited availability (e.g. a wet-lab in natural sciences). In all these cases, being able to generate a quality batch of query points without having to wait for previous labels to be acquired can be extremely advantageous. The importance of the scenario is further evidenced by the volume of existing literature and ongoing research on this topic. See for example Hoi et al. (2006), Guo & Schuurmans (2008), Wei et al. (2015), Sener & Savarese (2018), Kirsch et al. (2019), and many more references therein, all of which are concerned with the batch AL setting. We thank R3 for making this point, and will add content motivating the batch AL setting to the paper.

**R3: Evaluated datasets** As a methodology paper, our goal is to demonstrate the usefulness of our proposed method in a broad range of scenarios. We performed experiments on several small- and large-scale regression and classification datasets. While we agree with R3 that the datasets evaluated in the experimental section do not necessarily reflect real-world scenarios, the experimental protocols we used resemble benchmarks from multiple important (batch) AL papers (e.g., Hernandez-Lobato & Adams, 2015; Gal et al., 2017; Sener & Savarese, 2018). In fact, our work goes beyond what is typically tractable with Bayesian approaches (e.g., *cifar10* and *year*), demonstrating the usefulness and scalability of our method. Since the performance of the method is independent of labelling cost, we argue it is sufficient to use real-world settings as motivating examples (see discussion above) and leave specific applications to future work.

[Meta-Review · NeurIPS 2019]

The reviewers concluded that this paper offers valuable contributions and addresses an important practical problem. The authors also conduct an extensive set of experiments. However, I would strongly encourage the authors to consider the baseline suggested by R2, which is simple and requested by all other reviewers. A second baseline worth considering is to phantasize evaluation, akin what folks to do in Bayesian optmization. While this would not be attractive computationally, it encourages diverse batches. It would be interesting to see how the proposed method compares to this. Finally,a discussion of the computational complexity and/or limitations of current and competing techniques would also improve the paper.